# Programmable nanowrinkle-induced room-temperature exciton localization in monolayer WSe$_2$

Emanuil S. Yanev [1], Thomas P. Darlington [1], Sophia A. Ladyzhets [1], Matthew C. Strasbourg [2], Chiara Trovatello[1], Song Liu[1], Daniel A. Rhodes [1,3], Kobi Hall [1], Aditya Sinha[1], Nicholas J. Borys [2] ✉, James C. Hone [1] ✉ & P. James Schuck [1] ✉

Localized states in two-dimensional (2D) transition metal dichalcogenides (TMDCs) have been the subject of intense study, driven by potential applications in quantum information science. Despite the rapidly growing knowledge surrounding these emitters, their microscopic nature is still not fully understood, limiting their production and application. Motivated by this challenge, and by recent theoretical and experimental evidence showing that nanowrinkles generate strain-localized room-temperature emitters, we demonstrate a method to intentionally induce wrinkles with collections of stressors, showing that long-range wrinkle direction and position are controllable with patterned array design. Nano-photoluminescence (nano-PL) imaging combined with detailed strain modeling based on measured wrinkle topography establishes a correlation between wrinkle properties, particularly shear strain, and localized exciton emission. Beyond the array-induced wrinkles, nano-PL spatial maps further reveal that the strain environment around individual stressors is heterogeneous due to the presence of fine wrinkles that are less deterministic. At cryogenic temperatures, antibunched emission is observed, confirming that the nanocone-induced strain is sufficiently large for the formation of quantum emitters. At 300 K, detailed nanoscale hyperspectral images uncover a wide range of low-energy emission peaks originating from the fine wrinkles, and show that the states can be tightly confined to regions <10 nm, even in ambient conditions. These results establish a promising potential route towards realizing room temperature quantum emission in 2D TMDC systems.

Since their discovery in 2015[1–6], single photon emitters (SPEs) in monolayer transition metal dichalcogenides (TMDCs) have garnered significant interest from the scientific community for a wide range of applications in emerging quantum technologies[7,8]. While substantial progress has been made towards improving both the positioning and purity of SPEs, their underlying nature remains unresolved. Researchers have shown that strain often plays a key role in activating quantum emitters in these materials at low temperatures, and is

---

[1]Department of Mechanical Engineering, Columbia University, New York, NY, USA. [2]Department of Physics, Montana State University, Bozeman, MT, USA. [3]Department of Materials Science and Engineering, University of Wisconsin-Madison, Madison, WI, USA. ✉e-mail: nicholas.borys@montana.edu; jh2228@columbia.edu; p.j.schuck@columbia.edu

particularly effective in funneling and localizing excitons[9–17]. More recently, atomistic simulations have predicted that nanoscale wrinkles with extremely sharp radii of curvature can provide localization potentials large enough to create quantum-dot-like states even at room temperature[18,19]. Extensive experimental efforts have been directed at applying local strain to 2D sheets by, for example, using holes[5], pillars[20–28], particles[29,30], bubbles[19,31–34], indentations[35], tips[17,36,37], elastic polymers[38], pyramids on cantilevers[39], gaps[40–45], and edges[12,46–48] as passive and active stressors, all of which have enabled the quasi-deterministic positioning (sub-micron) of local emitting states at cryogenic temperatures. However, the properties and effects of the wrinkles induced by such structures have not been well studied, largely due to the lack of spatial resolution necessary to resolve them.

Here, we show that the engineered properties of stressor array lattices provide an additional knob for controlling exciton localization. By exploiting the fact that wrinkles preferentially form when a 2D layer conforms to non-planar topography, we show that the symmetry of the stressor array guides the dominant orientation of the wrinkles. Using far-field and near-field photoluminescence (nano-PL), we map out the optoelectronic properties of the induced wrinkles from micro to nano length scales, showing enhanced low-energy emission from these areas. Specifically, the localized room-temperature emission coincides with fine nanowrinkles in monolayer (1L) tungsten diselenide (WSe$_2$) sheets emanating from the nanocone stressors. Correlating the nano-PL hyperspectral maps with nanoscale strain maps of the same wrinkles, which are obtained from our AFM topography-based strain modeling, reveals correlations between localized exciton (LX) emission and strain. At cryogenic temperatures, the emergence of SPEs from localized exciton states in the strained system is observed. Coupled with the greatly reduced defect density in the flux-grown[49] material used, these results provide strong evidence that nanowrinkles are critical for quantum emission in strain-based TMDC systems.

## Results

### Wrinkle system overview

Using lithographically defined topographical features to induce strain in 2D materials has become commonplace[7]. Here, we utilize plasmonic nanocones with tunable tip radii (SI Fig. 1) as our local stressors. This stressor design is motivated by two considerations: (i) previous work has linked the observed emission of quantum-dot-like emitters in TMDCs at elevated temperatures with local plasmonic enhancement[19,23,24]; and (ii) for patterned pillar structures, deeply-confined emission tends to be observed around their rims or edges[47] rather than at a specific point. The cones used in this work increase localization by bending the 2D material about a much smaller point, thereby raising the magnitude of induced strain and facilitating the formation of nanowrinkles possessing sharp curvatures (Fig. 1).

Examples of nanocone-induced wrinkling in 1L-WSe$_2$ can be seen in the tilted scanning electron microscope (SEM) images of Fig. 1b. The 1L-WSe$_2$ conforms strongly to these cones without obvious signs of puncturing. The nanocone arrays were fabricated from substrate-supported Au films through a sequence of lithography, Al$_2$O$_3$ deposition, and argon ion milling, as illustrated in Fig. 1c. Exfoliated monolayer flakes of WSe$_2$ were then transferred onto the nanocones using polycaprolactone (PCL) on a polydimethylsiloxane (PDMS) stamp. A more in-depth discussion can be found in the methods section.

### Structural characterization

SEM micrographs of isolated cones reveal the narrow widths of some wrinkles, while also clearly showing that wrinkles can extend for hundreds of nanometers from their point of origin. To investigate the effect that introducing a periodic array of cones has on wrinkle formation, 1L-WSe$_2$ was transferred onto cone arrays with a pitch of 500 nm, in both square and triangular arrangements. AFM (Fig. 2) and optical (Fig. 3) imaging reveals extended networks of interconnected wrinkles bridging many of the lattice sites (Fig. 2a, d), which we term "array wrinkles." Differences in the likelihood of forming these connections are observed between the samples, as shown by the probability distributions in SI Fig. 2. Beyond the array symmetry, we speculate that several aspects influence the formation and direction of such array wrinkles, including the lattice spacing, cone height, uncontrolled residual strain from the WSe$_2$ transfer process, and alignment of the array symmetry relative to the crystallographic symmetry of the material; more in-depth investigations will be required for quantifying the role of each of these factors independently and is beyond the scope of the present study.

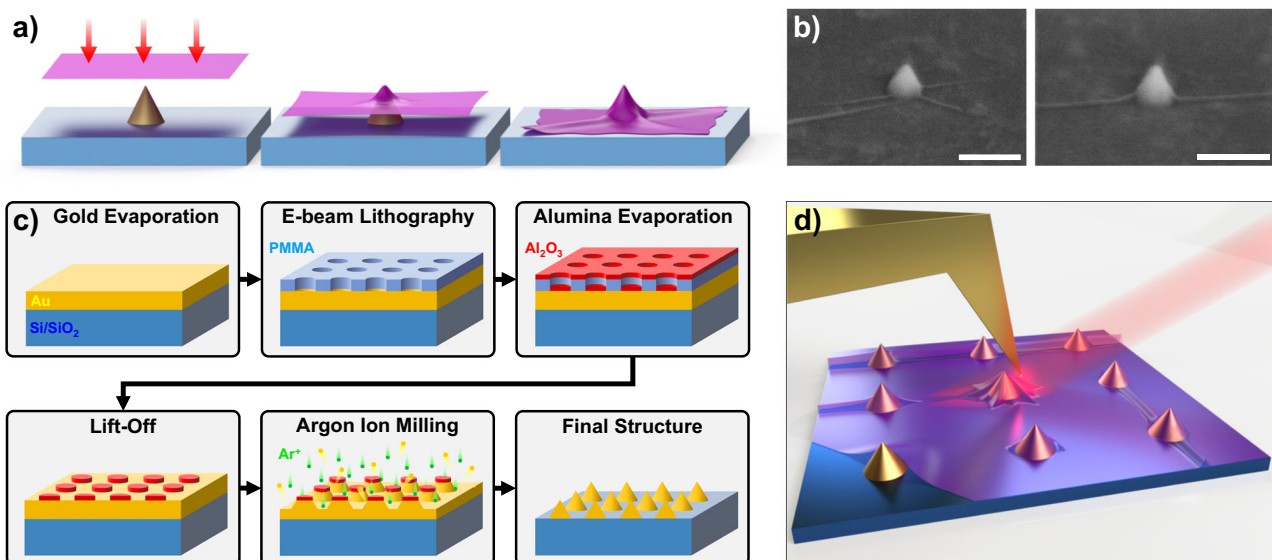

**Fig. 1 | 2D materials on nanocones. a** Cartoon of wrinkle formation when a sheet conforms to a conical feature. **b** SEM micrographs of wrinkles in monolayer WSe$_2$ on top of isolated nanocones. Scale bars are 200 nm. **c** Fabrication flow for the production of nanocone arrays. Abbreviations used: Si/SiO$_2$ silicon substrate with oxide layer, Au gold, PMMA poly(methyl methacrylate), Al$_2$O$_3$ aluminum oxide, Ar+ argon ions. **d** An illustration of the near-field measurement depicting a gold-coated AFM probe scanning over wrinkles in a sheet of WSe$_2$ on top of an array of nanocones. The tip and sample are illuminated from the side with a laser, which creates a local hotspot directly underneath the apex. (Drawing not to scale).

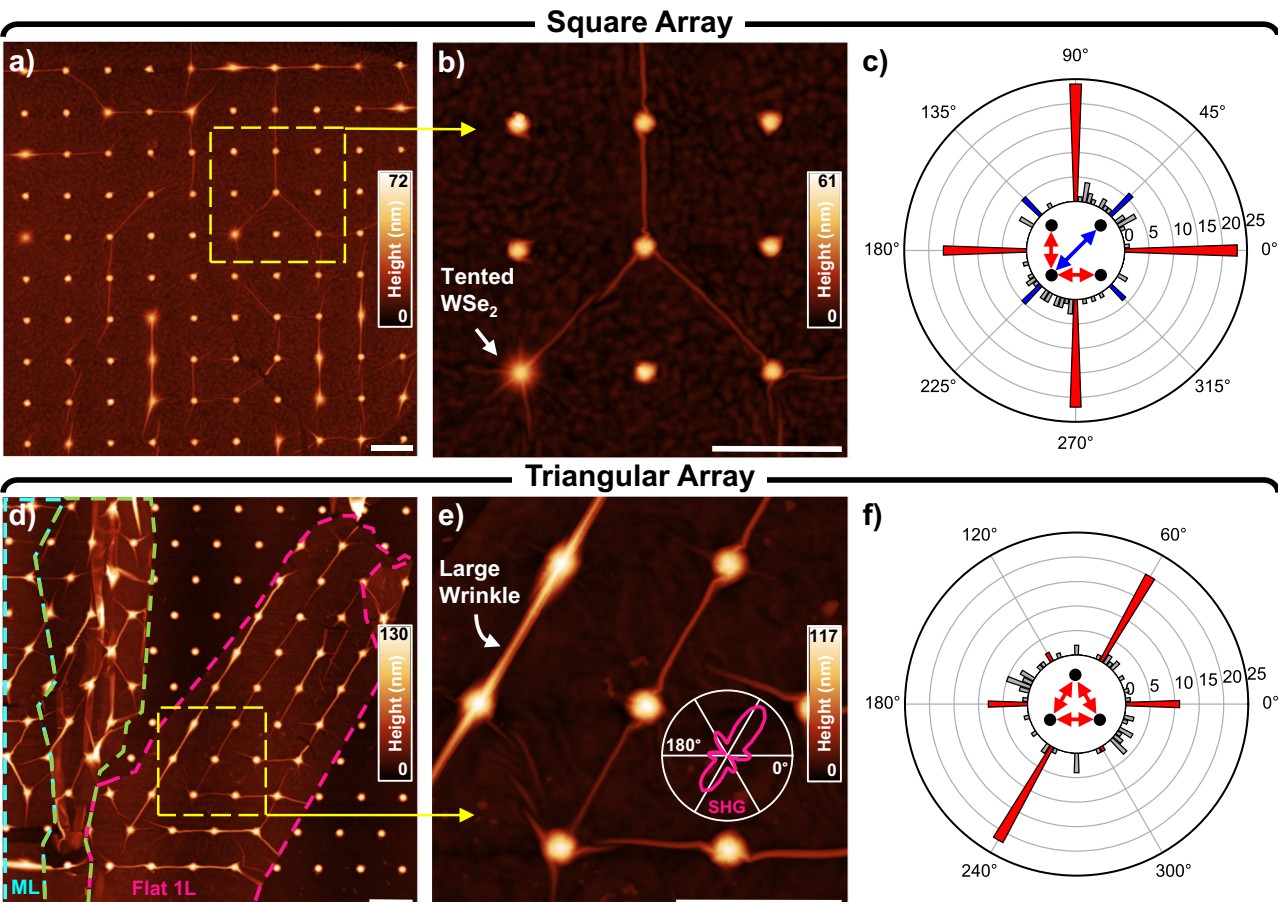

**Fig. 2 | AFM characterization of array wrinkles on substrates with different lattice symmetries. a** Winkles in a monolayer of WSe₂ on a square array of nanocones. **b** Detail of yellow square in (**a**) exhibiting both strong and weak conformity of the WSe₂ to the cone. **c** A polar histogram of wrinkle directions in (**a**), with the bins corresponding to lattice directions colored red. **d** WSe₂ on a triangular array of nanocones, with the flat monolayer region outlined in pink, the multilayer in cyan, and a transition zone where the monolayer crumpled up during transfer in green. **e** Detail of yellow square in (**d**) showing kinking of some wrinkles as they spiral around cones. The measured SHG response from the monolayer is overlaid in pink and indicates that the crystallographic axes are closely aligned to the nanocone array. **f** A polar histogram of wrinkle directions in (**d**), with the bins corresponding to lattice directions colored red. Only wrinkles in the flat (pink) monolayer region were counted. All scale bars are 500 nm.

A careful tally of the wrinkles and their angular orientation illustrates that wrinkles are indeed guided by the chosen array symmetry (Fig. 2c, f). For the square lattice, the predominant directions are vertical and horizontal, corresponding to wrinkle formation between nearest-neighbor sites, as previously shown in graphene[50,51]. Although less common, diagonal wrinkles between second-nearest-neighbors are also present at elevated levels (Fig. 2c; highlighted in blue). In the case of our triangular array, we find that the 60°/240° direction is favored, as is the 0°/180° direction to a lesser extent. Interestingly, we observe a smaller amount of wrinkling along 120°/300°. We attribute this apparent imbalance to the shape of the flake, which is long and narrow, as well as to directional strain unintentionally imparted during the transfer process, as supported by polarized second harmonic generation (SHG) measurements[52] on the 1 L region of this sample (Fig. 2e, overlay). We note that while only wrinkles in the flat monolayer area outlined in pink (Fig. 2d) were considered for the histogram, the adjoining multilayer region extending to the left also contains a significant number of wrinkles with angular orientations similar to those in the monolayer. The distribution for this area can be seen in SI Fig. 3, along with a multilayer region of the square lattice sample. Additionally, SI Fig. 4 demonstrates that an increase in the nanocone period can suppress the formation of array wrinkles, as expected[50]. More generally, our results show that the array degree of freedom can be a useful tuning knob for strain-based 2D systems.

High resolution AFM imaging also allows us to observe finer wrinkle-related features. Specifically, good conformal coverage can give rise to wrinkles that spiral and twist as the sheet contorts around the cone (Fig. 2b, e, center). Detailed scans of these structures reveal wrinkles with out-of-plane radii on the order of the AFM tip (nominally 2 nm) and in-plane meandering curves of 5–10 nm (SI Fig. 5). Such bends may further increase spatial confinement for optical excitations. Additionally, rather than complete conformal coating, in some cases a larger wrinkle (Fig. 2e, top left) or tent-like structure (Fig. 2b, bottom left) is formed, often containing many smaller creases. The optical properties of such nanoscale wrinkles are highlighted below (Figs. 4 and 5).

### Far-field photoluminescence

To investigate the exciton emission properties of wrinkles induced by the nanocone stressor arrays, we first performed far-field hyperspectral PL mapping on the square array sample using 633 nm CW excitation at room temperature. Topography of the entire scanned region is shown with an inverted and truncated color scale to highlight the small monolayer wrinkles (Fig. 3a). Much larger wrinkles formed in the multilayer region are visible even with a diffraction-limited white-light microscope (SI Fig. 6). Typical far-field spectra from bright and dim regions of the sample are shown in panel (b). While only the primary exciton (PX) peak is apparent at room temperature, the insets

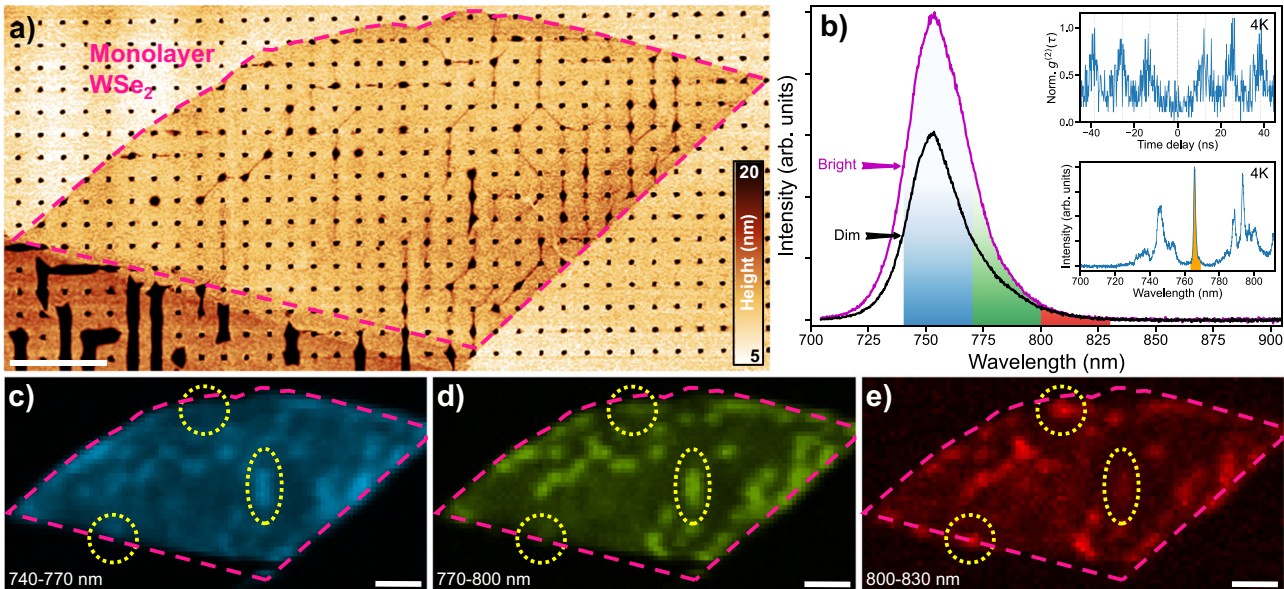

**Fig. 3 | Heterogeneous far-field optical properties. a** AFM topography of the confocal PL scan area, with the monolayer highlighted in pink. The dark lines are wrinkles between the cones. **b** Averaged spectra from a bright and dim region of the sample, corresponding to the yellow ellipse and featureless area to its left in (**d**). The two insets showcase low-temperature data from a similar wrinkle sample, demonstrating antibunched emission from a spectrally isolated narrow emitter (marked in orange). **c**–**e** Confocal PL maps showing the integrated intensity in the three correspondingly colored spectral windows in (**b**). The areas circled in yellow highlight the presence of localized states at different energies. All scale bars are 2 µm.

highlight narrow linewidths and antibunched emission from similar samples at 4 K. This low-temperature spectroscopy confirms the presence of SPEs in our nanocone/wrinkle system at cryogenic temperatures. Additionally, maps of PL intensity vary as a function of emission wavelength. Images of intensity generated by sweeping a 30 nm wide integration window from 740 nm to 830 nm show significant spatial inhomogeneity, particularly when the integration window is positioned on the low-energy tail of the PX peak. Example locations that exhibit strong spatial-spectral inhomogeneity on the filtered hyperspectral maps are highlighted in yellow in Fig. 3c–e. To help provide further insight, the total integrated intensity and spectral median are plotted in SI Fig. 7. The spatial variations suggest the existence of strain-localized low-energy LX states and motivate further investigation with nano-optical techniques[19,53].

### Near-field photoluminescence

Focusing on a wrinkled region around the center of the monolayer (encompassing the yellow ellipse in Fig. 3c–e), we collected hyperspectral nano-PL scans following the protocol detailed in methods. The AFM topography map (Fig. 4a) reveals many wrinkles in the area, and enhanced exciton emission over all wavelengths is seen primarily from the larger wrinkle near the middle (Fig. 4b, c). We note that the flat regions are dim, due primarily to PL quenching by a thin layer of Au remaining on the sample surface, which serves to suppress unwanted background signal from 1L-WSe₂ in these areas. In the large wrinkle, the 1L-WSe₂ is sufficiently separated from the substrate to overcome the quenching. Nano-PL emission maps are shown for the spectral window encompassing the (unstrained) PX peak (Fig. 4b, 740–770 nm) and for a lower-energy spectral window (Fig. 4c, 770–900 nm). In the low-energy window, emission is most intense in the wrinkle and in regions immediately surrounding the cones (Fig. 4c). In the higher energy window, while emission from the topmost cone apex is relatively dim, sizeable emission is present at the apexes of the other two cones (Fig. 4b). Rather than being fully quenched, Purcell-enhanced emission at the apexes of plasmonic nanocone antennas can shift the balance between radiative exciton recombination, nonradiative quenching, and funneling to lower-energy states, resulting in detectable PX PL

signal arising from such regions[9–16,54–57]. We note that cone-to-cone variations in enhancement and quenching are to be expected given nanoscale fabrication-related substrate heterogeneity.

Evidence of strain-localized low-energy exciton emission from different areas within the wrinkled region can be seen in the five sample spectra shown in Fig. 4d, corresponding to the locations marked with white arrows in Fig. 4c. These representative spectra illustrate the wide variety of emission lineshapes observed throughout the sample and infer the presence of a rich, heterogeneous, nanoscale landscape of strain imparted by topographical stressors.

Zooming in on the cone at the top of the array wrinkle that is highlighted in Fig. 4 reveals multiple small wrinkles or creases in the tent-like portion of the monolayer (Fig. 5a; as noted above, less PX emission is observed from this region). Several of these fine wrinkles are explicitly labeled to differentiate them from the larger array wrinkle that forms between neighboring cones. The dashed black or white circles in Fig. 5a–f demarcate the base of the cone underneath the tented WSe₂, whereas the green outlines in panels (c)–(f) are guides to the eye. Integrating the intensity over the full spectral range clearly shows bright nano-PL at the cone periphery (Fig. 5b), as previously discussed. In Fig. 5c, the spectral median was calculated for all pixels above a minimum threshold intensity. A large shift of ~80 nm (~160 meV) corresponds to the most intense regions of nano-PL. More generally, a comparison of panels (c) and (d) shows a clear correlation between emission energy and intensity within the strain-localized regions. This correlation is further quantified in SI Fig. 8. More direct comparison with strain is achieved by modeling local strain fields using a technique[58] based on AFM topography previously developed for nanobubbles. Generalizing the method to include wrinkles, we were able to estimate the normal (panel e) and shear (panel f) components of strain in the vicinity of the cone. These calculations reveal fairly large strains of ~2.5%, both normal and shear.

Multiple spectra of interest were extracted from the 10 numbered pixels in panel (b) and plotted in Fig. 5g. From these spectra, we observe that LX emission can span a broad range of energies due to the complex, heterogeneous strain environment around a single stressor, with multiple local maxima in strain serving to confine excitons in

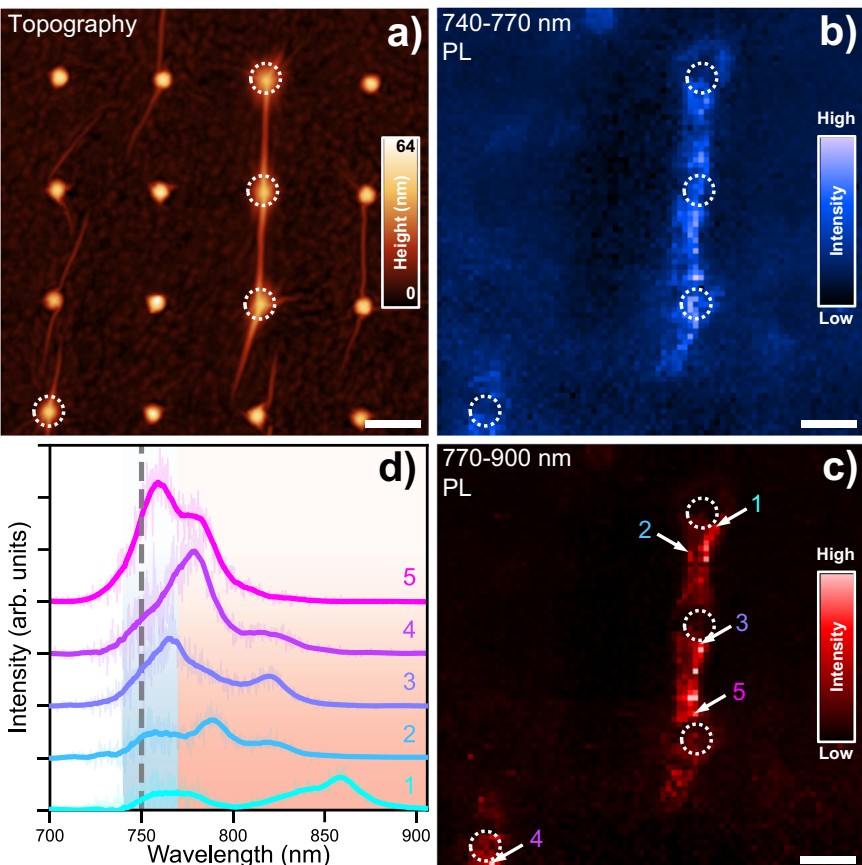

**Fig. 4 | Near-field optical investigation of a large array wrinkle. a** AFM topography of the nano-PL scanned region. White circles are a guide for the eye, marking the approximate location of the nanocone stressors. **b** Near-field intensity in the primary exciton spectral range of 740–770 nm. **c** Near-field intensity in the low-energy spectral range of 770–900 nm. **d** Select spectra, taken from the locations marked with white arrows in (**c**), showing a diverse assortment of low-energy states. The blue and red colored regions correspond to the integration ranges for (**b**) and (**c**), respectively. The bold solid lines are the result of a moving average overlaid on top of the raw data for clarity. All scale bars are 250 nm.

distinct but adjacent regions. The same behavior is observed in fine wrinkles around other cone locations, with LX peaks occurring at even longer wavelengths—out to, and potentially beyond, 900 nm (SI Fig. 9; long-wavelength detection is limited by detector bandwidth in our setup). Panel (h) shows 5 neighboring spectra originating from a line cut through one of the fine wrinkles. A prominent LX peak is only observed from the single 10 nm pixel at the middle of the line cut. This demonstrates the high degree of spatial confinement of excitonic states by the fine wrinkles, in good agreement with prior work[18,19]. Similar confinement can be seen at other locations in this data set (SI Fig. 10a, b), and others (SI Fig. 10c, d). Notably, power-dependent nano-PL measurements from wrinkles show a sublinear dependence of the low-energy LX emission on pump intensity (SI Fig. 11), providing further evidence of the localized nature of the states[28,59]. These results are consistent with previous reports on the effects of local strain modulation[12,17,34,37], with the nanoscale control and correlations reported here additionally highlighting the potential for substrate-induced nanowrinkles as sources of room-temperature strain-localized exciton state emission, as well as the challenges in deterministically positioning such states with nanoscale precision.

## Discussion

In summary, placing 2D materials on top of nanocone substrates is an effective way to create strained wrinkles with preferential orientations prescribed by the symmetry of the patterned features. In addition to array wrinkles that connect neighboring cones, fine wrinkles are formed by topographic stressors. Although their shapes and directions are more stochastic, their ability to strongly confine excitons is much greater, potentially leading to SPEs at room temperature. Using hyperspectral nano-PL imaging, we observe a wide gamut of low-energy peaks—spanning a range of over 235 meV—around a single nanocone stressor. This heterogeneity highlights the fact that the strain environment around artificial structures is often highly inhomogeneous and may need to be better controlled for future applications that seek uniform emitter properties. The emission from these states is extremely localized in some cases, reaching resolution-limited confinement in scans with step sizes of 2–10 nm. Given their spatial extent at room temperature, it is possible that these sites could exhibit single photon emission, as previously suggested by theory[18] and bolstered by observations of anti-bunched emission from the system at cryogenic temperatures. This work adds a new element of control while also elucidating the role of underlying nanoscale heterogeneity, yielding a deeper understanding of strain on localized exciton behavior, which will be crucial for deterministically positioning and tuning localized emitters for nano- and quantum-photonics applications.

## Methods

### Sample fabrication

Arrays of nanoscale gold cones were fabricated through a process adapted from Schäfer et al.[60]. First, a 5 nm thick titanium adhesion layer was evaporated onto a Si/SiO$_2$ substrate, followed by 100 nm of gold. The thickness of these layers determines the maximum height of the final structures. PMMA and MMA copolymer were spun on top of the gold to form a bilayer electron-beam resist stack of around 110 nm

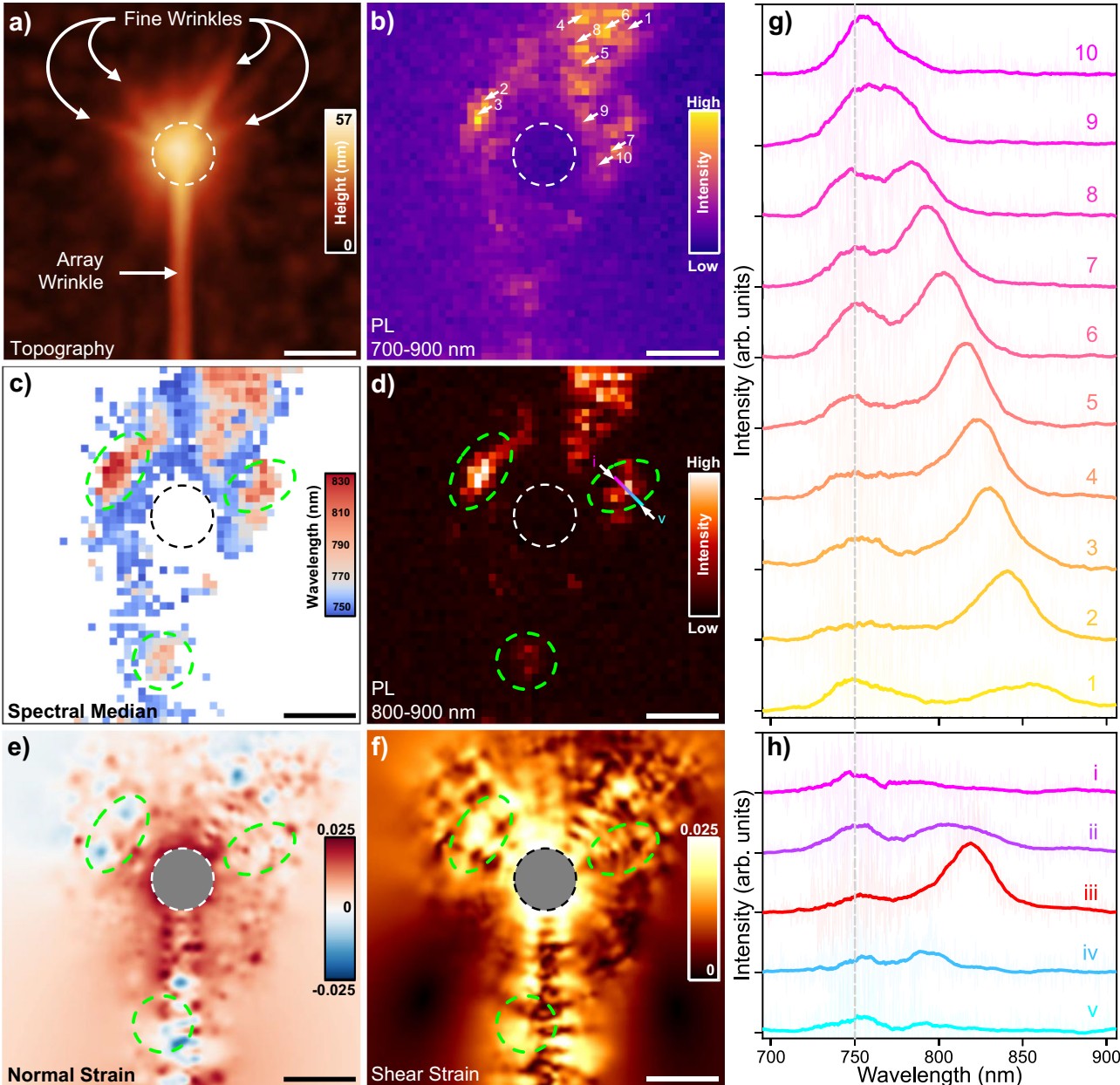

**Fig. 5 | Near-field optical properties of fine nanowrinkles. a** AFM topography of the nano-PL scan region showing very small wrinkles radiating out from the top of the cone. The dashed white circle marks the approximate size and location of the nanocone base. **b** Nano-PL integrated intensity from 700–900 nm, with spectra from the numbered pixels plotted in (**g**). **c** A spatial map of the spectral median, where pixels with an integrated intensity below some threshold value were excluded for clarity. The green dashed markings are guides to aid comparison with subsequent panels. **d** Nano-PL from localized states in the range of 800–900 nm. Spectra along the colored line are plotted in (**h**). **e** Normal and **f** shear strains calculated from AFM topography exhibit maximum values of ~2.5% away from the cone center, which has been excluded from the images due to ambiguities in the strain state at the tip apexes and to highlight the fine wrinkles. **g** Spectra taken from the locations marked with white arrows in (**b**), clearly showing that LX states span a spectral range of >100 nm below the primary exciton, presumably due to varying amounts of strain. It should be noted that the vertical arrangement of spectra in this panel was arbitrarily chosen based on emission energy and does not correspond to any spatial ordering. **h** Spectra taken from a cut through one of the fine wrinkles. The location is marked with a colored line in panel (**d**), and the start and finish are labeled with lowercase roman numerals i and v, respectively. The LX peak is well confined to a single 10 nm pixel in the middle. All scale bars are 100 nm.

thick. This resist layer was subsequently exposed and developed with a mixture of IPA and MIBK to form an array of circular wells. Aluminum oxide was then evaporated through the openings, leaving behind thin disks on the gold surface after lift-off in acetone. Finally, argon ion milling was performed at normal incidence until the aluminum oxide hard mask was fully consumed. Due to the angular dependence of both the sputtering yield and sidewall redeposition[61], this process produces features with sloped edges. Since circular masks were used, the result after milling was an array of cones with tip radii as low as 2 nm in optimized cases.

Flakes of 2D materials were then dry transferred onto the cone arrays. High quality TMDC crystals were first grown in house via the flux synthesis method[49], and then mechanically exfoliated onto Si/SiO$_2$ substrates using the well-known Scotch tape technique[62]. Monolayers were then directly picked up using a PCL stamp. Suhan Son et al. provide a detailed discussion of PCL stamp preparation and usage[63]. In short, PCL was spun to a uniform thickness and placed on top of a PDMS square attached to a glass slide, in similar fashion to stacking methods with other polymers. Flakes were brought into contact with the PCL by heating the substrate from 50 °C to 59 °C, and then picked

up by cooling down to 40 °C. The monolayers were then transferred onto the nanocone substrates by increasing the temperature above PCL's melting point of 60 °C and slowly lifting the stamp. Finally, the polymer was removed by submerging the samples in hot (70–80 °C) acetone for 1 h.

### Optical measurements

**Near-field.** Room-temperature PL was collected from the nanocone samples using a near-field scanning optical microscope (OmegaScope; Horiba Scientific) in conjunction with a Raman spectrometer (LabRAM HR Evolution; Horiba Scientific). Light from a 633 nm excitation laser was passed through a microscope objective (100x, 0.7 NA) at an angle of 65° relative to the surface normal and focused onto the apex of a gold coated AFM probe (AppNano Omni™ TERS-NC-Au). Collection of the sample luminescence was also accomplished with the same objective. Regular non-contact AFM scans were performed to locate the regions of interest, after which a special scanning modality (Dual-Spec; Horiba Scientific) was used to collect the PL. In this mode, the tip is brought into and out of contact with the sample at each pixel and a spectrum is collected at both tip positions. Due to a strong distance dependence, the spectra collected with the tip retracted away from the surface are almost entirely devoid of near-field contributions and serve as a good approximation of the far-field background. These were subsequently subtracted from the spectra obtained with the tip in contact to produce the hyperspectral nano-PL maps in Figs. 4 and 5, and SI Figs. 9 and 10.

**Second Harmonic Generation (SHG).** The crystal orientation of the WSe$_2$ sample was characterized using polarization-resolved SHG. As a laser source, we used a Coherent Chameleon Ultra II laser emitting tunable pulses from 680 nm to 1080 nm with temporal duration of ~150 fs, repetition rate of 80 MHz and average power of 4 W. For the SHG measurement, the pump wavelength was set to 1000 nm and spectrally filtered using two short pass filters at 1100 nm and two long pass filters at 900 nm. The polarization of the pump was controlled by rotating a half waveplate. The pump power was attenuated to 1.2 mW and focused onto the sample using a 50 x objective with a numerical aperture of 0.95. The 500 nm SHG signal emitted from the sample was collected in reflection geometry through the same objective and isolated from the pump using two short pass filters at 550 nm and 600 nm. An analyzer locked to the pump polarization was used to condition the signal before it passed through a grating spectrometer (Princeton Instruments SpectraPro HRS-300) and impinged on an EMCCD (Princeton Instruments ProEM). The polarization-dependent SHG was calibrated using a CVD grown MoS$_2$ monolayer flake with regular triangular shape as reference.

**Low-temperature optical measurements.** Photon antibunching was measured by conducting Hanbury-Brown and Twiss (HBT) interferometry at 4 K. In brief, a 1L-WSe$_2$-on-Au-nanostressor sample was cooled to 4 K in a liquid helium cryostat with optical access (Montana Instruments s50 Cryostation). Pulsed laser light (NKT Photonics SuperK EXTREME; 55 ps pulse width and spectrally filtered to 532 nm) was focused onto the surface of the sample using a home-built laser-scanning confocal microscope with a spatial resolution of ~1 μm. The photoluminescence spectra from the sample were recorded by using a grating spectrometer (Horiba iHR320) in conjunction with a spectroscopy CCD camera (Andor idus 416). For the antibunching measurements, the PL from a single narrowband emitter was spectrally isolated by using the grating spectrometer as a monochromator (Horiba iHR320). The filtered PL was relayed to a balanced beam splitter and detected using single-photon avalanche photodiodes. The second-order intensity correlation function (g2) was calculated by cross-correlating the time-dependent intensities on the arms of the HBT

interferometer. All PL and photon antibunching measurements were conducted with a laser CW-equivalent power of ~3 μW.

### Finite element strain modeling

The strain was calculated following the method described in reference[58] which models the monolayer TMDC as a classical plate. The Airy stress tensor is first calculated by solving the 2$^{nd}$ of the Foppl-von Karman equations using the Simple Finite Element Methods in Python[64]. The strain is then estimated from the Airy stress function via numerical differentiation.

## Data availability

All data generated or analyzed during this study, which support the plots within this paper and other findings of this study, are included in this published article and its Supplementary Information. Source data are provided with this paper.

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

## Acknowledgements

This work was supported by the National Science Foundation through award NSF DMR No. 2004437. T.P.D., J.C.H. and P.J.S. acknowledge support from Programmable Quantum Materials, an Energy Frontier Research Center funded by the US Department of Energy (DOE), Office of Science, Basic Energy Sciences (BES), under award DE-SC0019443. Nanostressor fabrication was partially supported through a US Department of Energy, Office of Science Graduate Student Research (SCGSR) award (E.Y.) and Honda Research Institute USA, Inc. The SCGSR-supported effort utilized the Nanofabrication facility of the Center for Functional Nanomaterials (CFN), which is a U.S. Department of Energy Office of Science User Facility, at Brookhaven National Laboratory under Contract No. DE-SC0012704. The authors acknowledge the use of facilities and instrumentation supported by NSF through the Columbia University, Columbia Nano Initiative, and the Materials Research Science and Engineering Center DMR-2011738. This work was performed in part at the Nanofabrication Facility at the Advanced Science Research Center at The Graduate Center of the City University of New York. C.T. acknowledges the European Union's Horizon Europe research and innovation programme under the Marie Skłodowska-Curie PIONEER HORIZON-MSCA-2021-PF-GF grant agreement No 101066108. N.J.B. also acknowledges the MonArk NSF Quantum Foundry supported by the National Science Foundation Q-AMASE-i program under NSF award No. DMR-1906383.

## Author contributions

P.J.S., J.C.H., N.J.B. and E.S.Y. conceived the nano-optical study. Experimental measurements were conducted by E.S.Y., M.C.S. and C.T. Substrate fabrication and sample preparation for these measurements was carried out by E.S.Y. High-quality TMDC crystals were provided by S.L. and D.A.R. The experimental data was analyzed by E.S.Y., M.C.S., C.T., N.J.B. and P.J.S. Strain calculations from AFM topography were carried out by T.P.D., S.A.L., K.H., and A.S. All authors contributed to the preparation of the manuscript.

## Competing interests

The authors declare no competing interests.
