## [Peer Review File · Nature Communications]

Reviewers' Comments:

Reviewer #1:

Remarks to the Author:

The manuscript describes a study of far-field and near-field photoluminescence (PL) of strain induced monolayer WSe₂ to demonstrate room-temperature exciton localization in the materials. This subject is timely and important for achieving room temperature single-photon emissions from semiconducting 2D materials. However, the experimental evidence of exciton localization in the manuscript is solely based on the emission wavelength (energy), which seems insufficient to confirm the presence of localized excitons.

1. Localized excitons usually refer to excitons bound to defects or nanoscale zero-dimensional traps, exhibiting distinct emission characteristics such as emission energy, lifetime, power saturation, and photon antibunching when compared to free exciton emissions (or primary exciton PX, as referred to in this manuscript). These localized excitons differ from the strain-induced bandgap modulation of PX excitons.

2. It is well-known that tensile strain lowers the bandgap and exciton energy in WSe₂ at a rate of ~ 60 meV/%. Based on their strain calculations in figures 5e and 5d, the PX peak can locally shift by ~ 150 meV, which is ~ 75 nm in terms of wavelength. This observation aligns well with the spectra shown in figures 5e and 5h. Consequently, it remains unclear whether the lower energy peaks in the manuscript originate from the localized excitons (LX) or strain-modulated primary excitons (PX). Therefore, the authors should provide additional evidence to support room temperature exciton localization.

3. The authors might be able to check power-dependent PL emission intensity, as the intensity of LX is saturating at high power while that of PX is linear. If the localized states are in a single exciton regime, the authors may also find photon antibunching.

4. The authors have also found narrow local strain modulation in a nanometer scale through wrinkled structures. However, there are previous reports on nanoscale local strain modulation that showed local reshifts and broadening of PX peaks (e.g. Nano Letters 23, 198 (2023), Nat. Commun., 13, 232 (2022), Sci. Adv., 7, 44 (2021), Nano Lett., 20, 9, 6791, (2020)), so the authors also need to clarify the novelties.

5. Have the authors observed any correlation between the direction of the transfer process and the direction of wrinkles?

Reviewer #2:

Remarks to the Author:

The manuscript entitled "Programmable Nanowrinkle-Induced Room-Temperature Exciton Localization in Monolayer WSe₂" by Yanev et al. reports on the fabrication of strained wrinkles in monolayer WSe₂ by placing the material on Au nanoconical substrates. They investigate the correlation between topographical stress factors and localized photoluminescence (PL) emission. While the study addresses an important area of research in the field of strain-induced emission in two-dimensional materials, which has gained significant attention recently, I have concerns regarding the novelty and scientific impact of the reported phenomena and characteristics, which are essential for publication in Nature Communications. At the very least, if the authors can demonstrate deterministic control of wrinkles or implementation of single photon emission through such control, resubmission may be considered. Therefore, I recommend the rejection of the manuscript with advice for significant revision and resubmission.

My major concerns regarding the work are as follows:

1. It is difficult to accurately identify the bilayer area in Figure 2d. I suggest the authors indicate this area clearly in the image. Additionally, it would be helpful to include images from other samples in the Supporting Information to judge the wrinkle formation angle (especially monolayer

vs. bilayer) and provide a distribution of angles.

2. For mechanically exfoliated samples, it is possible to predict the lattice direction based on the cut plane. Can the authors use this information to predict the correlation with the alignment of array symmetry? Furthermore, do other TMD materials exhibit similar trends?

3. In Figure 3b, it would be beneficial to show not only the averaged spectrum but also the overall distribution of PL spectra. Including spectra from representative points (e.g., yellow highlighted points and dark areas with small signals) in the confocal PL maps (c-e) would enhance our understanding of the PL spectra distribution.

4. In Figure 5, it is challenging to determine the correlation between Figures (c-d) and (e-f). It is unclear whether there is strain inside WSe₂ in the regions where localized emission appears. I suggest the authors mark the areas they want to highlight in the figure or provide further explanations to clearly identify these regions.

Reviewer #3:

Remarks to the Author:

Single-photon emitters formed by straining a two-dimensional TMDC are of great interest, but their origin remains to be explored. Single-photon emission from localized excitons formed in two-dimensional materials has been experimentally realized only at cryogenic temperatures. It would be very meaningful if this single-photon emitter could be realized at room temperature. Yanev et al. reported the optical properties of nanowinkles formed in WSe₂ monolayers. The authors fabricated gold nanocones to create nanowinkles and transferred WSe₂ monolayers onto them. The authors analyzed the orientation of the formed winkles according to the array of nanocone stressors and measured the PL in the near and far fields of the nanowinkles at room temperature. The researchers estimated the strain based on the experimental results of the nanowinkles. The results may have important implications for the development of single-photon sources operating at room temperature. This paper is recommended for publication in Nature Communications with the following revisions.

1) PL measurements at cryogenic temperatures are very useful in the study of excitons in two-dimensional materials. It would be great to show cryogenic PL measurements and scanning images of nanowinkles.

2) To realize room temperature single photon emission from nanowinkles, single photon emission from nanowinkles at cryogenic temperatures must first be confirmed. It would be nice to show single photon emission from nanowinkles at cryogenic temperatures.

3) Controlling the nanowinkle also seems to be an important issue. The authors observed the formation of nanowinkles on square and triangular lattice structures. Does the formation of nanowinkles depend on the lattice period of the stressor?

4) How does the crystal orientation of a two-dimensional material affect the formation of nanowinkles?

RESPONSES TO REVIEWER COMMENTS

Reviewer #1 (Remarks to the Author):

The manuscript describes a study of far-field and near-field photoluminescence (PL) of strain induced monolayer WSe₂ to demonstrate room-temperature exciton localization in the materials. This subject is timely and important for achieving room temperature single-photon emissions from semiconducting 2D materials.

Our reply:

We thank the reviewer for their thoughtful reading of our manuscript and for noting the timeliness and importance of the work.

However, the experimental evidence of exciton localization in the manuscript is solely based on the emission wavelength (energy), which seems insufficient to confirm the presence of localized excitons.

1. Localized excitons usually refer to excitons bound to defects or nanoscale zero-dimensional traps, exhibiting distinct emission characteristics such as emission energy, lifetime, power saturation, and photon antibunching when compared to free exciton emissions (or primary exciton PX, as referred to in this manuscript). These localized excitons differ from the strain-induced bandgap modulation of PX excitons.
2. It is well-known that tensile strain lowers the bandgap and exciton energy in WSe₂ at a rate of ~ 60 meV/%. Based on their strain calculations in figures 5e and 5d, the PX peak can locally shift by ~ 150 meV, which is ~ 75 nm in terms of wavelength. This observation aligns well with the spectra shown in figures 5e and 5h. Consequently, it remains unclear whether the lower energy peaks in the manuscript originate from the localized excitons (LX) or strain-modulated primary excitons (PX). Therefore, the authors should provide additional evidence to support room temperature exciton localization.

Our reply (to comments 1 and 2 since they are closely related):

We appreciate the reviewer's constructive comments on this topic, and agree that further evidence of exciton localization would go a long way to supporting our statements and clarifying the nature of the observed nanoscale emission.

To this end, we have now included **new experimental data**, showing (i) **power-dependent saturation** of the nano-PL intensity from localized array-wrinkle states (see response to comment 3 below) and (ii) low-temperature far-field **antibunching** from array wrinkle regions (see below).

One clarification that we feel is important: along with the spectral data that provided evidence of exciton localization, we had also demonstrated direct nanoscale visualization of localized emission at these energies via nano-PL hyperspectral imaging (Figs. 4 and 5 in previous and updated versions of the manuscript), providing further compelling real-space evidence of localized emitting states.

We agree with the reviewer about the need for additional clarification in our language regarding the localized exciton emission. In this case we are referring to excitons which are confined by highly localized strain gradients such as those previously reported in nanobubbles. To make this convention more explicit, we have changed “localized” to “strain-localized” wherever appropriate in the manuscript. We believe our use of “-localized” is appropriate here since our nano-PL images show that the emission originates from nanoscale spatial regions.

Relatedly, the reviewer is correct that the spectral shifts shown in the waterfall plots of Figs. 5g and h do correlate well with our calculated strain in those local regions and the known dependence of exciton emission on strain. A key feature to note is that each of these spectra is from a distinct nanoscale localized region, with the spectral properties of the low-energy feature changing (and frequently disappearing) over a length scale consistent with that of the strain gradients involved, indicating strain localization of the excitons.

Changes to the manuscript:

-Based on reviewer suggestions, we have performed **far-field low-temperature measurements on array wrinkle samples** that have also demonstrated room-T localized low-energy nano-PL emission in the same regions. We show that the low-T PL from wrinkled regions indeed exhibits distinct emission characteristics including **narrow low-energy emission lines and antibunching**, indicative of the creation of local quantum-dot-like states. This data is now included in the **updated Fig. 3, panel b**.

Figure 3. Heterogeneous far-field optical properties. (a) AFM topography of the confocal PL scan area, with the monolayer highlighted in pink. The dark lines are wrinkles between the cones. (b) Averaged spectra from a bright and dim region of the sample, corresponding to the yellow ellipse and featureless area to its left in (d). The two insets showcase low-temperature data from a similar wrinkle sample, demonstrating antibunched emission from a spectrally isolated narrow emitter (marked in orange). (c-e) Confocal PL maps showing the integrated intensity in the three correspondingly colored spectral windows in (b). The areas circled in yellow highlight the presence of localized states at different energies. All scale bars are 2 μm .

-We have performed power-dependent nano-PL measurements on low-energy exciton emission from array nanowrinkles, showing clear saturation behavior indicative of exciton localization (see response to comment 3 below, and **new SI Fig. 11**).

-We have changed “localized” to “strain-localized” where appropriate:

“Motivated by this challenge, and by recent theoretical and experimental evidence showing that nanowrinkles generate **strain**-localized room-temperature emitters, we demonstrate a method to intentionally induce wrinkles with collections of stressors...”

“The spatial variations suggest the existence of **strain**-localized low-energy states and motivate further investigation with nano-optical techniques^{18,48}.”

“The areas circled in yellow highlight the presence of **strain**-localized states at different energies. All scale bars are 2 μm .”

“Evidence of **strain**-localized low-energy exciton emission from different areas within the wrinkled region can be seen in the five sample spectra shown in Fig. 4d...”

“These results highlight the potential for substrate-induced nanowrinkles as sources of room-temperature **strain**-localized exciton state emission, as well as the challenges in deterministically positioning such states with nanoscale precision.”

“(d) Nano-PL from **strain**-localized states in the range of 800–900 nm.”

3. The authors might be able to check power-dependent PL emission intensity, as the intensity of LX is saturating at high power while that of PX is linear. If the localized states are in a single exciton regime, the authors may also find photon antibunching.

Our reply:

This is an excellent suggestion by the reviewer. Motivated by this comment, we spent time addressing several technical challenges and constraints in our experimental setup, which has now allowed us to perform near-field power-dependent PL measurements. We are pleased to include a **new SI Figure 11** with these results, which show clear saturation behavior from the localized low-energy states.

The power-dependence measurements are repeated for both increasing and decreasing powers, showing that sample drift is minor over the timescale of the measurement and cannot explain the observed sublinear trend (SI Fig. 11b). In our measurements, we collect the nano-PL hyperspectral data for when the tip is down, as well as from when the tip is retracted from the sample, which represents approximately the far-field background signal. When the tip is down, the low energy nano-PL becomes prominent, and its power-dependence is clearly sublinear. Whereas when the tip is up, the emission is dominated by the free excitons (PX) and we observe the expected linear scaling.

Changes to the manuscript:

-We have added new SI Fig. 11 and discussion describing the results in the main text.

SI Figure 11. Near-field power dependence of localized emitters. (a) Intensity as a function of incident power for integration regions around the primary exciton (725–775 nm, blue circles) and low-energy emitters (850–900 nm, red circles) in an array wrinkle. The tip location during collection is marked with a green “X” on the inset AFM map. The dashed lines are power law fits to the data which show linear (sublinear) scaling for the PX (LX) region. A half decade vertical offset has been applied to the red data points for easier comparison. (b) Total integrated intensity as a function of incident power with the tip in contact (green circles) and out of contact (purple circles) with an array wrinkle. The green “X” on the inset AFM map marks the location of interest. In order to check for drift, the power was swept up and then back down with minimal hysteresis observed from start to finish. Dashed lines are fits to a subset of the data comprising of the 3 highest powers (5 data points). The purple data points have been vertically offset by half a decade for clarity. All scale bars are 100 nm.

-Added main text discussion:

“Notably, power-dependent nano-PL measurements from wrinkles show a sublinear dependence of the low-energy LX emission on pump intensity (SI Fig. 11), providing further evidence of the localized nature of the states^{28,59}.”

4. The authors have also found narrow local strain modulation in a nanometer scale through wrinkled structures. However, there are previous reports on nanoscale local strain modulation that showed local reshifts and broadening of PX peaks (e.g. Nano Letters 23, 198 (2023), Nat. Commun., 13, 232 (2022), Sci. Adv., 7, 44 (2021), Nano Lett., 20, 9, 6791, (2020)), so the authors also need to clarify the novelties.

Our reply:

We agree that these reports include relevant demonstrations and have now included citations to these works. We thank the reviewer for reminding us of them (we regret missing them the first time) and for providing the opportunity to clarify our advances relative to previous work.

In general, our results are consistent with the observations in these reports. **The primary novelty and advance** offered by our work is the demonstration of controlled quasi-deterministic wrinkle formation and orientation within arrays (and hence formation and alignment of localized exciton states), utilizing the array degree of freedom, which to the best of our knowledge has not been investigated in earlier TMD studies.

Similar to reports 1 and 2 mentioned by the reviewer, we perform nano-optical studies of the strain localization. And similar to reports 3 and 4, we measure effects of strain-induced exciton funneling and LX energy tuning. *Beyond these reports*, we have added both the control of wrinkle formation and our ability to directly correlate nano-optical and strain mapping in these systems. Our new results deepen the understanding of nanoscale exciton behavior and expose key challenges that remain in achieving the ultimate goal of engineering fully deterministic arrays of quantum emitters in 2D semiconductors.

Changes to the manuscript:

-We have added citations to these works, as well as the following text:

“These results are consistent with previous reports on the effects of local strain modulation^{12,17,34,37}, with the nanoscale control and correlations reported here **additionally highlighting** the potential for substrate-induced nanowrinkles as sources of room-temperature **strain**-localized exciton state emission, as well as the challenges in deterministically positioning such states with nanoscale precision.”

5. Have the authors observed any correlation between the direction of the transfer process and the direction of wrinkles?

Our reply:

We share the reviewer's curiosity on this topic. In fact, we have observed some evidence of this correlation. But we feel it is premature to draw a definitive conclusion on this topic due to the difficulty in quantitatively tracking the directionality of transfer and applied strain. In particular, in our experience we find that contact fronts do not always propagate consistently in direction or velocity. This is true both on the way in and out during material pickups and transfers.

That said, our **new SHG measurements** of crystallographic properties establish a correlation between direction of induced residual strain in the TMD and the preferential formation of wrinkles. In particular, our polarized SHG 6-fold "flower" pattern in **updated Fig. 2e** is asymmetric with an elongated lobe in the direction that corresponds to the most common wrinkle direction observed on the sample. Previous work has shown that the orientation of the larger lobe in such measurements correlates with the direction of strain within the material [Mennel, et al., Nature Communications 9, 516 (2018)].

We also point out that in this sample, the SHG measurement shows that the WSe_2 crystallographic orientation is approximately aligned with the array. Together, these results are consistent with wrinkle formation and orientation that is favorably influenced by crystal-array alignment and directionally applied strain, though more detailed, systematic study is required to quantify the relationships.

Changes to the manuscript:

-We have updated Fig. 2e showing SHG measurements on the triangular array:

Figure 2. AFM characterization of array wrinkles on substrates with different lattice symmetries. (a) Wrinkles in a monolayer of WSe_2 on a square array of nanocones. (b) Detail of yellow square in (a) exhibiting both strong and weak conformity of the WSe_2 to the cone. (c) A polar histogram of wrinkle directions in (a), with the bins corresponding to lattice directions colored red. (d) WSe_2 on a triangular array of nanocones, with the flat monolayer region outlined in pink, the multilayer in cyan, and a transition zone where the monolayer crumpled up during transfer in green. (e) Detail of yellow square in (d) showing kinking of some wrinkles as they spiral around cones. The measured SHG response from the monolayer is overlaid in pink and indicates that the crystallographic axes are closely aligned to the nanocone array. (f) A polar histogram of wrinkle directions in (d), with the bins corresponding to lattice directions colored red. Only wrinkles in the flat (pink) monolayer region were counted. All scale bars are 500 nm.

Reviewer #2 (Remarks to the Author):

The manuscript entitled “Programmable Nanowrinkle-Induced Room-Temperature Exciton Localization in Monolayer WSe₂” by Yanev et al. reports on the fabrication of strained wrinkles in monolayer WSe₂ by placing the material on Au nanoconical substrates. They investigate the correlation between topographical stress factors and localized photoluminescence (PL) emission. While the study addresses an important area of research in the field of strain-induced emission in two-dimensional materials, which has gained significant attention recently, I have concerns regarding the novelty and scientific impact of the reported phenomena and characteristics, which are essential for publication in Nature Communications. At the very least, if the authors can demonstrate deterministic control of wrinkles or implementation of single photon emission through such control, resubmission may be considered. Therefore, I recommend the rejection of the manuscript with advice for significant revision and resubmission.

Our reply:

We are grateful to the reviewer for noting the importance of this line of research and for their constructive feedback and suggestions.

Based on these suggestions, we have now performed **new experiments** on cone-array wrinkle samples, showing (i) **power-dependent saturation** of the nano-PL intensity from localized array-wrinkle states, (ii) low-temperature far-field **antibunching** from array wrinkle regions, and (iii) **dependence of controlled array wrinkle formation** on stressor array lattice period (see below).

Changes to the manuscript:

-We have performed **far-field low-temperature measurements on array wrinkle samples** that have also demonstrated room-T localized low-energy nano-PL emission in the same regions (note that the far-field collection region is diffraction limited to ~1 μm in diameter and thus signal is collected from multiple wrinkles at once). We show that, when cooled down, the low-T PL from wrinkled regions indeed exhibits emission characteristics including **narrow low-energy emission lines and antibunching**, indicative of local quantum-dot-like states. This far-field data is now included in the **updated Fig. 3, panel b**.

Figure 3. Heterogeneous far-field optical properties. (a) AFM topography of the confocal PL scan area, with the monolayer highlighted in pink. The dark lines are wrinkles between the cones. (b) Averaged spectra from a bright and dim region of the sample, corresponding to the yellow ellipse and featureless area to its left in (d). The two insets showcase low-temperature data from a similar wrinkle sample, demonstrating antibunched emission from a spectrally isolated narrow emitter (marked in orange). (c-e) Confocal PL maps showing the integrated intensity in the three correspondingly colored spectral windows in (b). The areas circled in yellow highlight the presence of localized states at different energies. All scale bars are 2 μm .

-We have added new SI Fig. 11 highlighting new power-dependent nano-PL measurements on low-energy exciton emission from array wrinkles, showing clear saturation behavior indicative of exciton localization.

SI Figure 11. Near-field power dependence of localized emitters. (a) Intensity as a function of incident power for integration regions around the primary exciton (725–775 nm, blue circles) and low-energy emitters (850–900 nm, red circles) in an array wrinkle. The tip location during collection is marked with a green “X” on the inset AFM map. The dashed lines are power law fits to the data which show linear (sublinear) scaling for the PX (LX) region. A half decade vertical offset has been applied to the red data points for easier comparison. (b) Total integrated intensity as a function of incident power with the tip in contact (green circles) and out of contact (purple circles) with an array wrinkle. The green “X” on the inset AFM map marks the location of interest. In order to check for drift, the power was swept up and then back down with minimal hysteresis observed from start to finish. Dashed lines are fits to a subset of the data comprising of the 3 highest powers (5 data points). The purple data points have been vertically offset by half a decade for clarity. All scale bars are 100 nm.

-Added new SI Fig. 4, showing the absence of array wrinkle formation for large cone array lattice periods (1 μm and 2.5 μm in this case), highlighting how array period is a useful knob for **controlling wrinkle formation**:

SI Figure 4. Samples with increased array pitch. WSe₂ on (a) 1 μm and (b) 2.5 μm spaced nanocones. The pink dashed boundaries delineate monolayer regions, while the cyan areas are multilayer. The large horizontal stripe in (a) is a gold divider from the nanocone fabrication process. No array wrinkles are evident in these samples, highlighting the fact that lattice spacing is an important control parameter for large wrinkle formation. Further investigation is needed to properly quantify this effect.

-Added main text discussion:

“Notably, power-dependent nano-PL measurements from wrinkles show a sublinear dependence of the low-energy LX emission on pump intensity (SI Fig. 11), providing further evidence of the localized nature of the states^{28,59}.”

My major concerns regarding the work are as follows:

1. It is difficult to accurately identify the bilayer area in Figure 2d. I suggest the authors indicate this area clearly in the image. Additionally, it would be helpful to include images from other samples in the Supporting Information to judge the wrinkle formation angle (especially monolayer vs. bilayer) and provide a distribution of angles.

Our reply:

We appreciate the suggestion and have **added a cyan outline of the multilayer region in Fig. 2d**. We have also marked a “crumpled” monolayer region that is present in green, which is part of a larger crumpled region on the sample. At present, we have excluded this region from our analysis since most of its topography reflects unknown issues with poor transfer in that region rather than nanostressor array effects.

In addition, we have included **new SI Figure 3** to showcase the wrinkle angle distribution in multilayer regions. The distributions are similar to those from the monolayer regions (Fig. 2). **New polarized SHG data (overlaid in Fig. 2e)** collected from the triangular array sample shows that within measurement accuracy the WSe₂ crystal lattice is aligned with the array lattice vectors (as also illustrated by the clean monolayer edge near the right side of the Fig. 2d). Additionally, the SHG data show the presence of residual strain in this crystal along the 60°/240° direction, suggesting the additional strain

may contribute to the dominance of wrinkle formation in this direction. Overall, the histograms show that the engineered nanostressor arrays enable a new degree of control over wrinkle directionality.

Changes to the manuscript:

-We have updated panels (d) and (e) of Figure 2 to clearly identify the multilayer region and crystallographic orientation of the flake.

Figure 2. AFM characterization of array wrinkles on substrates with different lattice symmetries.

(a) Wrinkles in a monolayer of WSe_2 on a square array of nanocones. (b) Detail of yellow square in (a) exhibiting both strong and weak conformity of the WSe_2 to the cone. (c) A polar histogram of wrinkle directions in (a), with the bins corresponding to lattice directions colored red. (d) WSe_2 on a triangular array of nanocones, with the flat monolayer region outlined in pink, the multilayer in cyan, and a transition zone where the monolayer crumpled up during transfer in green. (e) Detail of yellow square in (d) showing kinking of some wrinkles as they spiral around cones. The measured SHG response from the monolayer is overlaid in pink and indicates that the crystallographic axes are closely aligned to the nanocone array. (f) A polar histogram of wrinkle directions in (d), with the bins corresponding to lattice directions colored red. Only wrinkles in the flat (pink) monolayer region were counted. All scale bars are 500 nm.

-We have **added new SI Figure 3** with wrinkle angle distributions in multilayer regions of WSe_2 .

SI Figure 3. Wrinkle formation in thicker regions of WSe_2 adjacent to the monolayers in Fig. 2 of the main text. (a) AFM topography of multilayer WSe_2 on a square lattice of nanocones. The pink shaded area is part of the monolayer, which is the focus of the main text (see Fig. 3 and SI Fig. 6 for a complete view). (b) A polar histogram of wrinkle directions in (a), excluding the shaded region. (c) AFM topography of multilayer WSe_2 on a triangular lattice of nanocones. The green shaded area is the transition zone marked in Fig. 2d of the main text. (d) A polar histogram of wrinkle directions in (c), excluding the shaded region. The bins corresponding to lattice directions are colored red in both histograms. All scale bars are $1 \mu\text{m}$.

-Modified main-text discussion:

“A careful tally of the wrinkles and their angular orientation illustrates that wrinkles are indeed guided by the chosen array symmetry (Fig. 2c,f). For the square lattice, the predominant directions are vertical and horizontal, corresponding to wrinkle formation between nearest-neighbor sites, as previously shown in graphene^{50,51}. Although less common, diagonal wrinkles between second-nearest-neighbors are also present at elevated levels (Fig. 2c; highlighted in blue). In the case of our triangular array, we find that the $60^\circ/240^\circ$ direction is favored, as is the $0^\circ/180^\circ$ direction to a lesser extent. Interestingly, we observe a smaller amount of wrinkling along $120^\circ/300^\circ$. We attribute this apparent imbalance to the shape of the flake, which is long and narrow, as well as to directional strain unintentionally imparted during the transfer process, as supported by polarized second harmonic generation measurements⁵² on this sample

(Fig. 2e, overlay). We note that while only wrinkles in the flat monolayer area outlined in pink (Fig. 2d) were considered for the histogram, the adjoining multilayer region extending to the left also contains a significant number of wrinkles with angular orientations similar to those in the monolayer. The distribution for this area can be seen in SI Fig. 3, along with a multilayer region of the square lattice sample. Additionally, SI Fig. 4 demonstrates that an increase in the nanocone period can suppress the formation of array wrinkles, as expected⁵⁰. More generally, our results show that the array degree of freedom can prove to be a useful tuning knob for strain-based 2D systems.”

-SHG measurement details added to Methods:

Second Harmonic Generation (SHG):

The crystal orientation of the WSe₂ sample was characterized using polarization-resolved SHG. As a laser source, we used a Coherent Chameleon Ultra II laser emitting tunable pulses from 680 nm to 1080 nm with temporal duration of ~ 150 fs, repetition rate of 80 MHz and average power of 4 W. For the SHG measurement, the pump wavelength was set to 1000 nm and spectrally filtered using two short pass filters at 1100 nm. The polarization of the pump was controlled by rotating a half waveplate. The pump power was attenuated to 3 mW and focused onto the sample using a 50x objective with a numerical aperture of 0.95. The 500 nm SHG signal emitted from the sample was collected through the same objective and isolated from the pump using two short pass filters at 550 nm and 600 nm. An analyzer locked to the pump polarization was used to condition the signal before it passed through a grating spectrometer (Princeton Instruments SpectraPro HRS-300) and impinged on an EMCCD (Princeton Instruments ProEM). The polarization was calibrated using a CVD grown MoS₂ monolayer flake with regular triangular shape as reference.

2. For mechanically exfoliated samples, it is possible to predict the lattice direction based on the cut plane. Can the authors use this information to predict the correlation with the alignment of array symmetry? Furthermore, do other TMD materials exhibit similar trends?

Our reply:

We agree with the reviewer and believe the cut plane visible in Fig. 2d points towards this correlation. As noted above, to further verify this, we performed **polarized SHG measurements** to independently determine crystallographic orientation of exfoliated flake and report on residual strain (see **Fig. 2e overlay**). As noted by the reviewer, the cut plane does accurately reflect the WSe₂ crystal orientation and shows that it is aligned to the triangular array in Fig. 2, providing evidence of potential correlation between wrinkle formation and monolayer crystal/nanostressor array alignment. The new SHG measurements also establish a correlation between direction of induced residual strain in the TMD and the preferential formation of wrinkles. Together, these results are consistent with wrinkle formation and orientation that is favorably influenced by crystal-array alignment and directionally applied strain, though more detailed, systematic study is required to quantify the relationships and isolate their contributions.

Re. other TMD materials: While we have concentrated on WSe₂ due to the large existing literature on strain-induced quantum emission within its monolayers, work by others indeed implies that array-guided wrinkling occurs in other TMDs as well. Specifically, ref. [*Nat. Commun.* **6**, (2015)], while primarily focusing on strain-induced “artificial atoms” in MoS₂, shows stressor-connected wrinkles in Fig. 1c, d. Beyond TMDs, similar trends have been reported previously in graphene, as noted in the main text refs. [*Nano Lett.* **14**, 5044–5051 (2014); *Appl. Phys. Express* **4**, 075102 (2011)].

Changes to the manuscript:

-We have updated Fig. 2e showing SHG measurements on the triangular array.

A copy of updated Fig. 2e is provided above in response to Comment #1.

3. In Figure 3b, it would be beneficial to show not only the averaged spectrum but also the overall distribution of PL spectra. Including spectra from representative points (e.g., yellow highlighted points and dark areas with small signals) in the confocal PL maps (c-e) would enhance our understanding of the PL spectra distribution.

Our reply:

We appreciate the reviewer’s suggestion and agree this would be beneficial to the reader. We have now replaced the averaged spectrum of the entire sample with averaged spectra from a smaller bright region highlighted by a yellow circle and from a dark region (see **new panel (b) in Fig. 3 and associated caption**, also shown below for reference). We note that the far-field spectra collected at each pixel in Fig. 2 are all dominated by the PX peak, with spectral signatures of the localized states hidden in the low-energy tails of that peak (i.e., isolated LX peaks are generally not discernable in the far-field measurements, as they are overwhelmed by PX emission from their surrounding regions). To help provide further insight, we have now also calculated the spectral median at each pixel and plotted it in **new SI Fig. 7** along with the total intensity. While the spectral median cannot discern between strained PX emission and LX states, it provides a straight-forward picture of the shifting energy landscape in the sample, averaged over a diffraction-limited collection volume.

Changes to the manuscript:

-Figure 3b has been updated to show spectra from bright and dim regions of the sample, and low temperature data showing antibunching from a narrow emitter has been inset. The **figure caption, discussion in the main text, and methods section have been modified** to reflect these changes.

Figure 3. Heterogeneous far-field optical properties. (a) AFM topography of the confocal PL scan area, with the monolayer highlighted in pink. The dark lines are wrinkles between the cones. (b) Averaged spectra from a bright and dim region of the sample, corresponding to the yellow ellipse and featureless area to its left in (d). The two insets showcase low-temperature data from a similar wrinkle sample, demonstrating antibunched emission from a spectrally isolated narrow emitter (marked in orange). (c-e) Confocal PL maps showing the integrated intensity in the three correspondingly colored spectral windows in (b). The areas circled in yellow highlight the presence of localized states at different energies. All scale bars are 2 μm .

-We have added new SI Fig. 7 showing the spectral median plot of the flake in Fig. 3:

SI Figure 7. Additional details of far-field emission. (a) AFM topography of the monolayer in Fig. 3, reproduced here to aid comparison. (b) Confocal PL map of the total integrated intensity showing spatially inhomogeneous emission. (c) The calculated spectral median for pixels above some threshold in (b). Many similar features can be identified across all three panels, demonstrating a clear link between structure, brightness, and emission energy. All scale bars are 2 μm .

-Modified main-text discussion:

“To investigate the exciton emission properties of wrinkles induced by the nanocone stressor arrays, we first performed far-field hyperspectral PL mapping on the square array sample using 633 nm CW excitation. Topography of the entire scanned region is shown with an inverted and truncated color scale to highlight the small monolayer wrinkles (Fig. 3a). Much larger wrinkles formed in the multilayer region are visible even with a diffraction-limited white-light microscope (SI Fig. 6). Typical far-field spectra from bright and dim regions of the sample are shown in panel (b). While only the primary exciton (PX) peak is apparent at room temperature, the insets highlight narrow

linewidths and antibunched emission from similar samples at 4K. This confirms the presence of SPEs in our nanocone/wrinkle system at cryogenic temperatures. Additionally, maps of PL intensity vary as a function of emission wavelength. Images of intensity generated by sweeping a 30 nm wide integration window from 740 nm to 830 nm show significant spatial inhomogeneity, particularly when the integration window is positioned on the low-energy tail of the PX peak. Example locations that exhibit strong spatial-spectral inhomogeneity on the filtered hyperspectral maps are highlighted in yellow in Fig. 3c-e. To help provide further insight, the total integrated intensity and spectral median are plotted in SI Fig. 7. The spatial variations suggest the existence of strain-localized low-energy LX states and motivate further investigation with nano-optical techniques^{19,53}. ”

-Low temperature measurement details added to Methods:

Low-temperature optical measurements:

Photon antibunching was measured by conducting Hanbury-Brown and Twiss (HBT) interferometry at 4 K. In brief, a 1L-WSe₂ on Au nanostressor sample was cooled to 4 K in a liquid helium cryostat with optical access (Montana Instruments s50 Cryostation), pulsed laser light (NKT Photonics SuperK EXTREME; 55 ps pulse width and spectrally filtered to 532 nm) was focused onto the surface of the sample using a home-built laser-scanning confocal microscope with a spatial resolution of approximately 1 μm , and photoluminescence (PL) from a single emission feature was spectrally isolated using the monochromator exit port of a grating spectrometer (Horiba iHR320). The filtered PL was then relayed to a balanced beam splitter and detected using single-photon avalanche photodiodes. The second-order intensity correlation function (g_2) was calculated by cross-correlating the time-dependent intensities on the arms of the HBT interferometer.

PL spectra were measured by using a grating spectrometer (Horiba iHR320) in conjunction with a spectroscopy CCD camera (Andor idus 416). All PL and photon antibunching measurements were conducted with a laser power of approximately 3 μW .

4. In Figure 5, it is challenging to determine the correlation between Figures (c-d) and (e-f). It is unclear whether there is strain inside WSe₂ in the regions where localized emission appears. I suggest the authors mark the areas they want to highlight in the figure or provide further explanations to clearly identify these regions.

Our reply:

We appreciate the reviewer’s suggestion and have added additional annotations to guide readers.

Changes to the manuscript:

-We have updated panels c–f of Fig. 5 to highlight several regions of interest.

Figure 5. Near-field optical properties of fine nanowrinkles. (a) AFM topography of the nano-PL scan region showing very small wrinkles radiating out from the top of the cone. The dashed white circle marks the approximate size and location of the nanocone base. (b) Nano-PL integrated intensity from 700–900 nm, with spectra from the numbered pixels plotted in (g). (c) A spatial map of the spectral median, where pixels with an integrated intensity below some threshold value were excluded for clarity. The green dashed markings are guides to aid comparison with subsequent panels. (d) Nano-PL from localized states in the range of 800–900 nm. Spectra along the colored line are plotted in (h). (e) Normal and (f) shear strains calculated from AFM topography exhibit maximum values of $\sim 2.5\%$ away from the cone center, which has been excluded from the images due to ambiguities in the strain state at the tip apexes and to highlight the fine wrinkles. (g) Spectra taken from the locations marked with white arrows in (b), clearly showing that LX states span a spectral range of more than 100 nm below the primary exciton, presumably due to varying amounts of strain. It should be noted that the vertical arrangement of spectra in this panel was arbitrarily chosen based on emission energy and does not correspond to any spatial ordering. (h) Spectra taken from a cut through one of the fine wrinkles. The location is marked with a colored line in panel (d), and the start and finish are labeled with lowercase roman numerals i and v, respectively. The LX peak is well confined to a single 10 nm pixel in the middle. All scale bars are 100 nm.

Reviewer #3 (Remarks to the Author):

Single-photon emitters formed by straining a two-dimensional TMDC are of great interest, but their origin remains to be explored. Single-photon emission from localized excitons formed in two-dimensional materials has been experimentally realized only at cryogenic temperatures. It would be very meaningful if this single-photon emitter could be realized at room temperature.

Yanev et al. reported the optical properties of nanowinkles formed in WSe₂ monolayers. The authors fabricated gold nanocones to create nanowinkles and transferred WSe₂ monolayers onto them. The authors analyzed the orientation of the formed winkles according to the array of nanocone stressors and measured the PL in the near and far fields of the nanowinkles at room temperature. The researchers estimated the strain based on the experimental results of the nanowinkles. The results may have important implications for the development of single-photon sources operating at room temperature. This paper is recommended for publication in Nature Communications with the following revisions.

Our reply:

We are grateful to the reviewer for the encouraging words and suggestions, and for noting the important potential implications of our work.

1. PL measurements at cryogenic temperatures are very useful in the study of excitons in two-dimensional materials. It would be great to show cryogenic PL measurements and scanning images of nanowinkles.

Our reply:

We agree that low temperature measurements are quite useful, and we have now performed scanning PL imaging at 4K on cone-array samples (see example below) as well as g(2) photon correlation (antibunching) measurements (see response to comment #2 below). At cryogenic temperatures, we are limited to far-field diffraction limited resolution. Since the main focus of the current manuscript is high-resolution room-temperature imaging, we are aiming for a future work consisting of detailed low temperature characterization, analysis and quantitative correlation with nano-PL.

Figure R1: AFM (a) and low-temperature (4 K) scanning micro-photoluminescence images (b) from 1L-WSe₂ array wrinkles on a triangular nanocone array substrate. Single-photon emission was measured from this region. The antibunching curve and low-temperature PL spectrum is shown in Fig. 3b inset.

- To realize room temperature single photon emission from nanowinkles, single photon emission from nanowinkles at cryogenic temperatures must first be confirmed. It would be nice to show single photon emission from nanowinkles at cryogenic temperatures.

Our reply:

This is indeed an important check, and we have now included it as an inset to updated Figure 3b, showing clear antibunching at 4K to address this point.

Changes to the manuscript:

-Figure 3b has been modified to include low temperature PL and $g^{(2)}$ data.

Figure 3. Heterogeneous far-field optical properties. (a) AFM topography of the confocal PL scan area, with the monolayer highlighted in pink. The dark lines are wrinkles between the cones. (b) Averaged spectra from a bright and dim region of the sample, corresponding to the yellow ellipse and featureless area to its left in (d). The two insets showcase low-temperature data from a similar wrinkle sample, demonstrating antibunched emission from a spectrally isolated narrow emitter (marked in orange). (c-e) Confocal PL maps showing the integrated intensity in the three correspondingly colored spectral windows in (b). The areas circled in yellow highlight the presence of localized states at different energies. All scale bars are 2 μm .

- Controlling the nanowinkle also seems to be an important issue. The authors observed the formation of nanowinkles on square and triangular lattice structures. Does the formation of nanowinkles depend on the lattice period of the stressor?

Our reply:

This is an excellent point and question. Demonstrating the significance of wrinkles in stressor systems is the crux of this study, with the implication that substantial efforts will need to be undertaken to properly harness their potential and mitigate any detrimental effects. Strategies for doing so have yet to be investigated rigorously, but our work shows that the lattice period is definitely a parameter that impacts the formation of array wrinkles. We have observed the suppression of interconnecting wrinkles in samples with larger cone spacing, and **a new figure has been added to the supplementary information** to show several examples of this effect.

Changes to the manuscript:

-**SI Figure 4 has been added** to show WSe_2 flakes on arrays with larger spacing, in which negligible array wrinkle formation is observed.

SI Figure 4. Samples with increased array pitch. WSe_2 on (a) $1\ \mu\text{m}$ and (b) $2.5\ \mu\text{m}$ spaced nanocones. The pink dashed boundaries delineate monolayer regions, while the cyan areas are multilayer. The large horizontal stripe in (a) is a gold divider from the nanocone fabrication process. No array wrinkles are evident in these samples, highlighting the fact that lattice spacing is an important control parameter for large wrinkle formation. Further investigation is needed to properly quantify this effect.

4. How does the crystal orientation of a two-dimensional material affect the formation of nanowrinkles?

Our reply:

We had the same question. To address this, we performed **polarized SHG measurements** to independently determine crystallographic orientation of exfoliated flake and report on residual strain (see **Fig. 2e overlay**). We found that the cut plane visible in Fig. 2d accurately reflects the WSe_2 crystal orientation and shows that it is aligned to the triangular array in Fig. 2, providing evidence of potential correlation between wrinkle formation and monolayer crystal/nanostressor array alignment. The new SHG measurements also establish a correlation between direction of induced residual strain in the TMD and the preferential formation of wrinkles. Together, these results are consistent with wrinkle formation and orientation that is favorably influenced

by crystal-array alignment and directionally applied strain, though more detailed, systematic study is required to quantify the relationships and isolate their contributions.

Changes to the manuscript:

-We have updated Fig. 2e showing SHG measurements on the triangular array.

Figure 2. AFM characterization of array wrinkles on substrates with different lattice symmetries. (a) Wrinkles in a monolayer of WSe_2 on a square array of nanocones. (b) Detail of yellow square in (a) exhibiting both strong and weak conformity of the WSe_2 to the cone. (c) A polar histogram of wrinkle directions in (a), with the bins corresponding to lattice directions colored red. (d) WSe_2 on a triangular array of nanocones, with the flat monolayer region outlined in pink, the multilayer in cyan, and a transition zone where the monolayer crumpled up during transfer in green. (e) Detail of yellow square in (d) showing kinking of some wrinkles as they spiral around cones. The measured SHG response from the monolayer is overlaid in pink and indicates that the crystallographic axes are closely aligned to the nanocone array. (f) A polar histogram of wrinkle directions in (d), with the bins corresponding to lattice directions colored red. Only wrinkles in the flat (pink) monolayer region were counted. All scale bars are 500 nm.

Reviewers' Comments:

Reviewer #1:

Remarks to the Author:

The authors have included additional power-saturation and antibunching data to support the presence of localized excitations at the wrinkled positions. I would recommend publication in Nature Communications.

Reviewer #2:

Remarks to the Author:

I appreciate the author's efforts to answer the questions. Revised manuscript with the added figures of low-temperature PL measurement and the added physical interpretations looks much more solid. I think the manuscript may be published in Nature communications.

Reviewer #3:

None

RESPONSES TO REVIEWER COMMENTS

Reviewer #1 (Remarks to the Author):

The authors have included additional power-saturation and antibunching data to support the presence of localized excitons at the wrinkled positions. I would recommend publication in Nature Communications.

Reviewer #2 (Remarks to the Author):

I appreciate the author's efforts to answer the questions. Revised manuscript with the added figures of low-temperature PL measurement and the added physical interpretations looks much more solid. I think the manuscript may be published in Nature communications.

Our reply:

We are grateful to the reviewers for their careful reading and constructive feedback, which has improved and strengthened our manuscript.